# Pulmonary Recruitment Prior to Intraoperative Multiple Pulmonary Ground-Glass Nodule Localization Increases the Localization Accuracy—A Retrospective Study

**DOI:** 10.3390/jcm12082998

**Published:** 2023-04-20

**Authors:** Yu Hsiang Wang, Pei Chin Su, Hsu Chih Huang, Kenneth Au, Frank Cheau Feng Lin, Chih Yi Chen, Ming Chih Chou, Jiun Yi Hsia

**Affiliations:** 1Division of Thoracic Surgery, Chung Shan Medical University Hospital, Taichung 402, Taiwan; 99311132tzu@gmail.com (Y.H.W.); keisu16@gmail.com (P.C.S.);; 2School of Medicine, Chung Shan Medical University, Taichung 402, Taiwan; 3Institute of Medicine, Chung Shan Medical University, Taichung 402, Taiwan

**Keywords:** hybrid computer tomography, pulmonary ground-glass nodule localization, video-assisted thoracic surgery, pulmonary recruitment

## Abstract

The standard treatment for early-stage lung cancer is complete tumor excision by limited resection of the lung. Preoperative localization is used before video-assisted thoracoscopic surgery (VATS) to improve the accuracy of pulmonary nodule excision. However, lung atelectasis and hypoxia resulting from controlling apnea during the localization procedure may affect the localization accuracy. Pre-procedural pulmonary recruitment may improve the respiratory mechanics and oxygenation during localization. In this study, we investigated the potential benefits of pre-localization pulmonary recruitment prior to pulmonary ground-glass nodule localization in a hybrid operating room. We hypothesized that pre-localization pulmonary recruitment would increase the localization accuracy, improve oxygenation, and prevent the need for re-inflation during the localization procedure. We retrospectively enrolled patients with multiple pulmonary nodule localizations before surgical intervention in our hybrid operating room. We compared the localization accuracy between patients who had undergone pre-procedure pulmonary recruitment and patients who had not. Saturation, re-inflation rate, apnea time, procedure-related pneumothorax, and procedure time were also recorded as secondary outcomes. The patients who had undergone pre-procedure recruitment had better saturation, shorter procedure time, and higher localization accuracy. The pre-procedure pulmonary recruitment maneuver was effective in increasing regional lung ventilation, leading to improved oxygenation and localization accuracy.

## 1. Introduction

Lung cancer was the leading cause of cancer-related deaths worldwide in 1990 [1]. The Taiwan Lung Cancer Screening in Never Smoker Trial (TALENT) revealed an approximately 2.6% lung cancer detection rate in Taiwan. In all, 96.5% of lung cancer cases were detected at stage 0–1. VATS has improved the long-term survival rate of lung cancer and provides a minimally invasive approach for the removal of pulmonary nodules [2]. However, early lung adenocarcinomas are difficult to palpate during VATS and unfeasible to pathologically evaluate from fine needle aspiration or biopsy [3]. Therefore, preoperative and intraoperative localization, using materials such as microcoils, hookwires, contrast media, dyes, and fluorescence tracers, is important for thoracic surgeons [4]. Several techniques have recently been proposed for preoperative or intraoperative pulmonary nodule localization, including “two-stage” localization using conventional computer tomography and “one-stage” localization in a hybrid operating room. Computer tomography-guided localization in a hybrid operating room followed by immediate resection has been shown to be a safe and efficient localization technique, reducing the “risky period” [5,6,7]. However, pulmonary localization in a hybrid operating room requires general anesthesia with endotracheal intubation and localization in the same end-inspiratory phase. During general anesthesia, lung atelectasis develops in 90% of the patients because of muscle paralysis, intravenous or inhalational anesthetics, and surgical position [8,9]. This lung atelectasis may progress rapidly during apnea in the end-inspiratory phase, resulting in a hypoxia event and interfering with the final localization accuracy. In a single pulmonary ground-glass nodule localization, the total apnea time is around 1–2 min, whereas in the localization of multiple pulmonary ground-glass nodules, the full localization procedure typically takes 3–5 min and may take longer depending on the location or the number of lesions. Appropriate ventilator settings and experienced localization teamwork are important issues for the procedure.

Prompt pulmonary recruitment in the perioperative phase has been proven to improve respiratory mechanics, chest wall compliance, and oxygenation in patients undergoing abdominal and cardiothoracic surgery [8,9,10]. In this clinical study, we sought to identify the benefits of pre-procedure pulmonary recruitment prior to pulmonary ground-glass nodule localization in a hybrid operating room. We hypothesized that pulmonary recruitment would improve oxygenation, prevent the need for re-inflation during the multiple pulmonary ground-glass nodules localization procedure, and further increase the localization accuracy.

## 2. Materials and Methods

We retrospectively reviewed all patients who underwent hybrid-operating-room multiple pulmonary nodule localization before surgical intervention at Chung Shan Medical University Hospital (CSMUH) between 1 January 2022 and 31 December 2022. The inclusion criteria were: (1) age higher than 18 years, (2) lung function examination classification as low risk on the American College of Chest Physicians (ACCP) perioperative surgical risk evaluation (both forced expiratory volume and diffusion capacity of carbon monoxide ≥ 80% predicted), and (3) multiple pulmonary ground-glass nodules for localization. In all, 23 patients underwent pre-procedure pulmonary recruitment, and 69 patients did not undergo recruitment. The decision to perform pre-procedure pulmonary recruitment was based on the surgeon’s and anesthesiologist’s preference and recorded in the anesthesia record. The details of the inclusion, exclusion, and classification of the patients in the study are shown in Figure 1. Perioperative parameters, including pulse oximetry, apnea time, re-inflation rate, pneumothorax, and total procedure time, were analyzed. This study was approved by the CSMUH Ethics Committee (Approval No. CS1-23036).

All patients underwent chest computed tomography with contrast and had a multidisciplinary team review their case prior to their operation. The size of the patients’ lesions was between 4 and 15 mm as shown on the CT prior to their operation. The solid portion of the pulmonary lesions in our study were less than 10 mm, and all the lesions were eligible for wedge resection as both diagnostic and therapeutic surgery based on the Japan Clinical Oncology Group (JCOG) 0804 trial [11].

Each patient underwent double-lumen endotracheal tube intubation in the supine position after induction. After positioning the patient into decubitus or 30° lateral tilt position, the anesthesiologist confirmed the double-lumen endotracheal tube position by bronchoscopy, ensuring pulmonary recruitment and ventilation safety. The pre-procedural recruitment maneuver was performed with sustained inspiration pressure at 30 cm H_2_O for 30 s by the anesthesia workstation (Aisys CS2™ anesthesia workstations, General Electric Healthcare, Chicago, IL, USA). After recruitment, the radiologist and the thoracic surgeon performed the pre-procedural scan for localization planning. The localization hookwires placement or the dye injection is dependent on the lesion-to-pleura distance. After localization, the patient immediately underwent the VATS operation in the hybrid operation room. When the resected nodule was diagnosed as adenocarcinoma in situ (AIS) or adenocarcinoma in situ at least (AIS at least) based on the frozen section, we closed the incision without additional lung resection. If invasive adenocarcinoma or another histological tumor was confirmed, we then checked the lesions size and considered performing lobectomy or segmentectomy as extensive resection. The entire algorithm of multiple pulmonary ground-glass nodule localization procedure and surgery is shown in Figure 2.

The hybrid localization system was implemented using C-arm CBCT (ARTIS Pheno^®^; Siemens Healthcare GmbH, Erlangen, Germany) and localization data (such as distance between the chest wall and the nodule and characteristics of the nodule) were obtained from the hybrid room software. Saturation was recorded using pulse oximetry via a sensor from the contralateral fingernail bed, displayed in the anesthesia workstations and the localization monitor. These parameters were based on previous literature and our anesthesiologist’s preference [12,13]. Ventilation during localization was performed in pressure control and volume guarantee mode (PCV-VG) or volume control mode (VCV). The tidal volume was set at 8–10 mL/kg of predicted body weight, and the fraction of inspired oxygen (FiO_2_) was set at 50%. Re-inflation was performed if pulse oximetry was <90% (SpO_2_ < 90). The apnea time was recorded by the anesthesiologist, starting at the time of localization and ending at the successful completion of the procedure without SpO_2_ < 90. The localization accuracy was determined using the hybrid computed tomography film, with the distance between the localization needles and the lesion not exceeding 5 mm in a three-dimensional view. The film was reviewed immediately after the procedure by two radiologists. The secondary outcomes were apnea time, saturation, re-inflation rate, procedure time, and pneumothorax rate.

The significance of differences between the two groups was evaluated using the χ^2^-test, the Fisher’s exact test, and the Mann–Whitney U test. The localization accuracy and the rate of re-inflation were calculated using multivariate analysis with logistic regression and odds ratios. Statistical significance was defined as *p* <  0.05. All statistical analyses were performed using Statistical Package for Social Sciences (IBM SPSS Statistics, version 25) and Excel software (Microsoft Corporation, Seattle, WA, USA).

## 3. Results

Ninety-two patients had multiple pulmonary ground-glass nodule localizations in the CSMUH hybrid operating room between 1 January 2022 and 31 December 2022 (Figure 1). Table 1 shows the characteristics of the patients included in the study. Most of our patients had two pulmonary lesions (*N* = 63, 68.5%), 18 patients had three pulmonary lesions (*N* = 18, 19.6%), and 11 patients (*N* = 11, 12.0%) had four pulmonary lesions. All pulmonary localization procedures were completed using a tidal volume of at least 8–10 mL/kg, based on our anesthesiologist’s previous experience. Twenty-one patients underwent contralateral wedge resection before multiple pulmonary nodule localizations (*N* = 21, 22.8%). Most pulmonary nodule localization procedures were completed in the pressure control and volume guarantee mode (*N* = 83, 77.2%), and few localization procedures were completed in the volume control mode (*N* = 9, 22.8%). There were no significant differences in age, sex, height, weight, body mass index, preoperative pulmonary function, functional vital capacity (FVC), forced expiratory volume in 1 s (FEV1), diffusing lung capacity for carbon monoxide (DLCO), lesion numbers, and other localization characteristics between the groups.

Most of the patients underwent wedge resection of the lung when frozen section analysis confirmed only AIS or adenocarcinoma of the lung with only a small proportion of solid part, whereas five patients (7.2%) in the non-recruitment group and two patients (8.7%) in the recruitment group underwent segmentectomy and lobectomy, respectively, after the pathologic diagnosis of lung adenocarcinoma or to achieve an adequate resection margin. There was no hookwires dislodgment induced by lobectomy in our study. There was no significant difference in surgery type between the groups.

The final pathology reports showed a primary pulmonary malignancy in 177 nodules, including invasive adenocarcinoma, AIS, and minimally invasive adenocarcinoma (MIA). There were only five secondary pulmonary malignancies, comprising metastasis from clear-cell renal cell carcinoma, nasopharyngeal carcinoma, and hepatocellular carcinoma (five nodules in three patients); all were treated with wedge resection. There were also 42 nodules composed of fibrosis or chronic inflammation. There was no significant difference in the pathology report between the groups.

Twenty-three patients underwent pulmonary recruitment prior to their nodule localization (25.0%), and 69 patients did not (75.0%). The average total procedure time across both groups was 20.0 min (Table 2). However, the recruitment group had significantly shorter procedure times (median 17 vs. 19 min, *p* = 0.043). Twenty-nine patients developed pneumothorax after nodule localization (*N* = 29, 31.5%), none of which required immediately thoracentesis due to iatrogenic pneumothorax for procedure completion. There was no significant difference in the apnea time (median 5.60 vs. 5.90 min, *p* = 0.658) or in the incidence of post-procedural pneumothorax (median 66.7% vs. 73.9, *p* = 0.610) between the two groups. However, the recruitment group had better saturation (92% vs. 88%, *p* = 0.016) and higher localization accuracy (82.6% vs. 55.1%, *p* = 0.025) at the end of the procedure.

A univariate logistic regression revealed that pre-procedure recruitment, history of lung operation, and BMI were all factors associated with localization accuracy (odds ratio, 0.23, 0.25, and 0.76, respectively). Pre-procedure recruitment, history of lung operation, and higher BMI were also significant predictors in the multivariate logistic analysis of localization accuracy (Wald’s test *p* = 0.024, 0.033, and 0.002, respectively, Table 3).

A univariate logistic regression revealed that pre-procedure recruitment and higher lesion number were both factors associated with the need for re-inflation during localization (odds ratio, 3.09 and 1.87, respectively). However, pre-procedure recruitment was the only significant factor in the multivariate logistic analysis for the need for re-inflation during localization (Wald’s test *p* = 0.043, Table 4).

## 4. Discussion

This study highlights the importance of pre-procedural pulmonary recruitment in the preoperative localization of partial solid ground-glass nodules of the lung in hybrid operation rooms. We believe this study to be the first of its kind that examines the application of pulmonary recruitment in hybrid operation room localization. Compared to traditional two-stage localization in CT rooms, one-stage localization in hybrid operation rooms by a thoracic surgeon can avoid the delayed treatment of localization-related complications. In addition, the localization performed by the thoracic surgeon can guide the resection planning based on the accuracy of the localization. This is a crucial step in the preoperative planning of lung cancer surgeries, as it allows the surgeon to accurately locate and remove a partially solid ground-glass nodule of the lung while minimizing the risk of damage to healthy tissue.

The accurate localization of these partially solid nodules is essential for a successful limited resection of the lung. However, during the localization process, regional atelectasis can occur, which can cause uneven lung collapse and interfere with the accuracy of the procedure. Research has shown that the amount of atelectasis is affected by the patient’s BMI and the apnea time and can be minimized by pulmonary recruitment [14,15]. In obese patients, an impeded diaphragm and muscle weakness limit the ability to hold their breath for an extended period, resulting in less apnea time and potentially accelerated atelectasis [16]. This physiology limitation may accelerate the atelectasis and result in a difficult localization accuracy. The number of lesions is another important factor influencing accuracy. In our experience, in single pulmonary ground-glass nodule localization, the apnea time is always <2 min, and most patients will not develop atelectasis interfering with the localization accuracy, as evidenced by the final hybrid CT film review. However, in multiple ground-glass nodules localization, the apnea time lasts approximately 5 to 6 min, resulting in increased oxygen consumption during the apnea phase and an increased risk of atelectasis formation. The region of collapsed lung can significantly alter the localization plan from the pre-scan planning and thereby affect the accuracy of the procedure. In patients undergoing a contralateral lung operation, the atelectasis between the bilateral lung fields is typically uneven and may interfere with the accuracy of the localization. However, we found that after the patients underwent pre-procedure recruitment, the atelectasis lung alveolar re-expansion kept the entire lung field open, causing the lung to collapse much more evenly during the localization procedure. As showed in Table 3, pre-procedural pulmonary recruitment had a positive effect on elevation accuracy, and a history of contralateral wedge resection and a high BMI interfered with the localization accuracy with statistical significance (*p* = 0.024, 0.033, and 0.002, respectively).

In the literature, various methods of pulmonary recruitment are discussed, including stepwise incremental PEEP, stepwise increase in tidal volume to a plateau pressure of 30 cm H_2_O, and sustained inflations to a peak inspiratory pressure of 30–40 cm H_2_O [13,17,18,19]. However, there remains uncertainty about which pulmonary recruitment method is best for use in general anesthesia, one-lung ventilation, or pre-localization recruitment in clinical practice. To determine the success of pulmonary recruitment, various physiological measurements such as arterial partial oxygen pressure (PaO2) plus arterial partial pressure of carbon dioxide (PaCO_2_) greater than 400 mmHg or electrical impedance tomography may be used [20,21]. However, in general anesthesia practice, these measurements may not always be practical. As a result, a sustained inspiration pressure at 30 cm H_2_O for 30 s by the anesthesia workstation may be chosen as a pulmonary recruitment method due to its time efficiency and effectiveness. An inappropriate position of the double-lumen endotracheal tube during pulmonary recruitment may lead to the overdistension of one lobe and lung injury, which could have severe consequences for the patient. Therefore, proper positioning of the double-lumen endotracheal tube should always be ensured before the initiation of any pulmonary recruitment technique.

In this study, we defined localization accuracy as 5 mm in three-dimensional post-procedure CT, despite several studies proposing considerably different definitions of the accuracy of pulmonary ground-glass nodule localization [22,23]. The more precise localization not only increased the lung parenchymal preservation, but also applied the safety margin much more easily. Most of the partially solid ground-glass nodules of the lung were around 6–10 mm in size in our study. The distance between the localization needles and the lesion did not exceed 5 mm, approximately equal to the half of the tumor size, making it easy to apply the surgical stapler for the wedge resection of the lung in our surgical experience. In our study, we chose localization hookwires and patent dye as the localization materials, although some comparative studies showed that hookwires have a lower success rate compared with Lipiodol localization and microcoils localization. Some hookwires dislodgment or migration can occur after localization and may require extensive lung parenchymal resection or lobectomy of the lung for hookwires removal [4,24,25]. We choose a 20-gauge Chiba needle (Hakko Co., Naganogen, Japan) as the vector to deliver the localization hookwires. Once the hybrid room CT confirmed the Chiba needle hit the target of localization, the hookwires were delivered carefully.

Hypoxia during the apnea phase of the localization was another issue in this study. Hypoxia is the leading cause of anesthesia-related mortality, and patients with hypoxia require immediate oxygenation [26,27]. A previous observational study found that mild hypoxia (SpO_2_ 86–90%) lasting from 2 to 30 min developed in more than half of the patients who underwent surgery [28]. Perioperative hypoxia events increase the incidence of cardiac complications and delirium [29,30]. In an in vivo rat model, local hypoxia resulting from lung atelectasis also induced an inflammatory response [31]. Perioperative hypoxia events are associated with obesity, age, chronic obstructive pulmonary disease, interstitial lung disease, and history of lung surgery, especially lobectomy [32,33]. A previous study found only a mild vital capacity change after wedge resection, which was recovered after 12 months [34]. In our study, 39 (*N* = 39, 42.4%) patients developed desaturation (SpO_2_ < 90%) during the apnea phase of the localization procedure and needed re-inflation. The pre-procedural pulmonary recruitment group had better saturation and fewer desaturation events during nodule localization (SpO_2_ 92% vs. 88%, re-inflation 26.1% vs. 35.9%). In multivariate logistic regression analysis, only pre-procedural recruitment was a significant predictor of the need for re-inflation (odds ratio 2.99, Wald’s test, *p* = 0.043). A history of contralateral wedge resection, lesion number, BMI, and ventilator mode were not associated with hypoxia and re-inflation rate. None of our patients had adverse effects after the hypoxia events.

One solution to apnea phase-related hypoxia is to conduct the localization procedure without any respiratory hold. However, as inspiration and expiration continue, in some electromagnetic navigation-guided localization software, the target motion between inspiration and expiration will be recorded and may increase from 5 mm to 18 mm [23]. These respiratory motions maybe unremarkable in the traditional CT room but may prevent an adequate image for pulmonary nodule localization due to severe motion artifacts. This contrasts with the “two-stage” preoperative CT-guided pulmonary nodule localization. Another solution to apnea phase-related hypoxia is to provide high inspiratory fraction oxygenation 2–3 min before the procedure for reduction of the alveolar nitrogen fraction and elevation of alveolar oxygenation [35]. However, a high alveolar oxygenation may lead to a rapid absorption atelectasis, which may cause lung atelectasis within 2 min [36,37,38]. After the onset of apnea, alveolar oxygen is consumed at 250 mL per minute, while carbon dioxide is excreted at 20 mL per minute [39]. Therefore, we regard pre-procedure pulmonary recruitment as the best way to prevent hypoxic events during the localization procedure.

Pre-procedure pulmonary recruitment is useful in maintaining lung opening and has been used to reduce postoperative pulmonary complications and acute respiratory distress syndrome for several decades [12,40]. However, no study has discussed the relationship between pulmonary recruitment and pulmonary nodule localization. Our study is the first to observe both the technical and the physiological benefits of pre-procedure pulmonary recruitment in pulmonary nodule localization in a hybrid operating room. The pre-procedure pulmonary recruitment re-expanded the collapsed alveoli and improved chest wall compliance before the hybrid computer tomography scan, which prevented hypoxic events during the procedure.

This study has several limitations. First, we included only relatively healthy patients to observe the effect of recruitment on localization accuracy, patient oxygenation, and re-inflation rate. The physiological benefits of pre-procedure pulmonary recruitment may be greater in patients with cardiopulmonary disease and morbidity-related obesity. Only two patients with morbid obesity (BMI, 31 and 30, based on the World Health Organization definition) were included in our study, which may underestimate the effect of pre-procedure pulmonary recruitment in patients with morbid obesity. Future studies should focus on evaluating the effects of pulmonary recruitment in a broader range of patient populations. Second, the study was not randomized, and pre-procedure pulmonary recruitment was performed based on the preference and experience of the anesthesiologist and surgeon without exact inclusion criteria. While the patient characteristics were reviewed, there may have been unmeasured confounding variables that could have affected the results. A randomized controlled trial would provide more convincing evidence of the benefits of pre-procedure pulmonary recruitment. Another comparative study was conducted in our hospital to analyze the processing of multiple ground-glass nodules in the lungs of morbidly obese patients within a hybrid operation room. The study also considered various risk factors for analysis. The decision making for whether to use pre-procedure pulmonary recruitment may be made clearer by further study. Third, the study was retrospective, and the patient population undergoing pre-procedure pulmonary recruitment was not balanced with the non-recruitment group. An allocation bias may have influenced the results, and a prospective study with a well-designed protocol would be necessary to confirm the benefits of pre-procedure pulmonary recruitment. Fourth, we did not record pulmonary recruitment hemodynamics. Some studies have reported recruitment-induced hypotension, which may cause morbidities and mortality [41]. During recruitment, bilateral hyperinflation of the lung and high transpulmonary pressure promptly restricted the venous return, resulting in hypotension. In our study, none of the patients developed severe hypotension during the recruitment procedure. We also included only patients classified as at low risk on the ACCP perioperative surgical risk evaluation, and no cardiovascular patients were included for the study. This study only provides preliminary evidence of the benefits of pre-procedure pulmonary recruitment in relatively healthy patients undergoing multiple partially solid ground-glass nodules of the lung localization in a hybrid operation room. The limitations of the study should be taken into consideration, and future studies should evaluate the effects of pulmonary recruitment in a more diverse patient population, using a randomized controlled design and assessing the safety of the procedure in high-risk patients.

## 5. Conclusions

In this retrospective observational study, the pre-procedure pulmonary recruitment maneuver was effective in increasing regional lung ventilation, leading to improved localization accuracy and patient oxygenation. The patient BMI, the pre-procedure pulmonary recruitment maneuver, and any history of contralateral lung wedge resection were all significant predictors of localization accuracy. Our study demonstrates the efficacy and feasibility of pulmonary recruitment prior to multiple pulmonary ground-glass nodule localization in the hybrid operating room. Further randomized control studies are needed to confirm our observations and determine the decision criteria in the future.

## Figures and Tables

**Figure 1 jcm-12-02998-f001:**
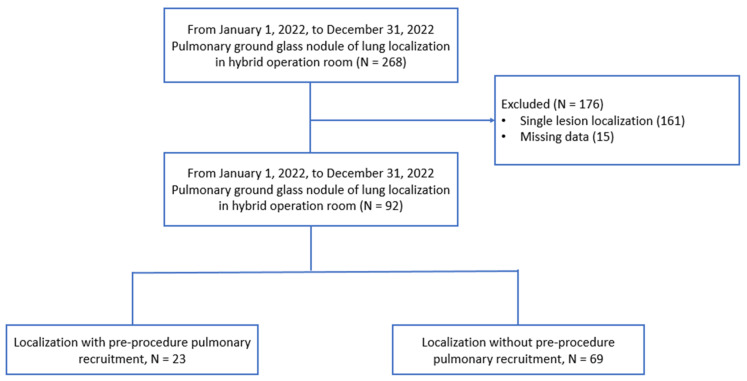
Flowchart of patient classification after their initial inclusion in the study.

**Figure 2 jcm-12-02998-f002:**
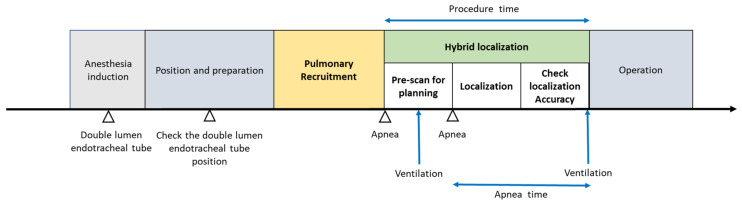
Phases of the multiple pulmonary ground-glass nodule localization and tumor excision operation.

**Table 1 jcm-12-02998-t001:** Characteristics of the study patients.

	Non-Recruitment Group (*N* = 69)	Recruitment Group (*N* = 23)	*p* Value
Age, median (IQR)	53 (47–63)	57 (49–66)	0.281
Gender			^†^ 0.778
Female	52 (75.4%)	18 (78.3%)	
Male	17 (25.6%)	5 (21.7%)	
Height, median (IQR)	160.0 (156.0–166.0)	160.0 (155.0–163.0)	0.857
Weight, median (IQR)	63.0 (55.0–70.0)	57.0 (52.0–69.0)	0.328
BMI	24.20 (22.20–26.20)	22.80 (20.30–26.00)	0.229
Preoperative lung function			
FVC	102 (94–110)	103 (96–108)	0.921
FEV1	97 (88–107)	103 (91–108)	0.195
DLCO	94 (85–102)	99 (90–104)	0.116
History of lung contralateral wedge resection	17 (24.6%)	4 (17.4%)	^†^ 0.473
Ventilation mode			^†^ 0.441
VCV	8 (11.6%)	1 (4.3%)	
PCVVG	61 (88.4%)	22 (95.7%)	
Tidal volume during procedure Median (IQR)	550 (525–600)	550 (500–600)	0.281
Lesions			^†^ 0.394
2 lesions	47 (68.1%)	16 (69.6%)	
3 lesions	12 (17.4%)	6 (26.1%)	
4 lesions	10 (14.5%)	1 (4.3%)	
Size (mm)	6.90 (5.70–8.30)	6.50 (5.50–8.20)	0.850
Dye/hook localization			^†^ 0.146
Dye localization	26 (37.7%)	13 (56.5%)	
Hook localization	43 (62.3%)	10 (43.5%)	
Depth, median (IQR)	70.0 (55.0–80.0)	65.0 (51.7–75.0)	0.691
Puncture times			^†^ 0.326
1 time	64 (92.8%)	23 (100%)	
>1 time	5 (7.2%)	0	
Extent of resection			^†^ >0.999
Wedge resection	64 (92.8%)	21 (91.3%)	
Segmentectomy ^†^ Wedge resection	2 (2.9%)	1 (4.3%)	
Lobectomy ^†^ Wedge resection	3 (4.3%)	1 (4.3%)	
Diagnosis			^†^ 0.591
Invasive adenocarcinoma	6 (3.5%)	3 (5.6%)	
Adenocarcinoma in situ (AIS)	81 (47.6%)	21 (38.9%)	
Minimally invasive adenocarcinoma (MIA)	51 (30.0%)	15 (18.5%)	
Secondary pulmonary malignancy	3 (1.8%)	2 (3.7%)	
Benign lesion	29 (17.1%)	13 (24.1%)	

Min, minutes; IQR, interquartile range; BMI, body mass index; FVC, forced vital capacity; FEV1, forced expiratory volume in one second; DLCO, diffusion capacity of carbon monoxide; VCV: volume control mode; PCV-VG: pressure control and volume guarantee mode. ^†^ Fisher’s exact test/Chi square test. Mann–Whitney U test.

**Table 2 jcm-12-02998-t002:** Comparisons between the recruitment and non-recruitment groups.

Outcome Setting	Non-Recruitment Group	Recruitment Group	*p* Value
	(*N* = 69)	(*N* = 23)	
Apnea time	5.60 (4.60–6.80)	5.90 (4.80–6.30)	0.658
(Min, Median, IQR)			
SpO_2_ (%, Median, IQR)	88.0 (84.0–93.0)	94.0 (87.0–97.0)	0.016
Re-inflation			^†^ 0.033
No need for inflation	36 (52.2%)	17 (73.9%)	
Need for inflation	33 (35.9%)	6 (26.1%)	
Procedure time	19.0 (15.0–24.0)	17.0 (14.0–19.0)	0.043
(Min, Median, IQR)			
Pneumothorax			^†^ 0.610
Pneumothorax	46 (66.7%)	17 (73.9%)	
No pneumothorax	23 (33.3%)	6 (26.1%)	
Accuracy (5 mm)	38 (55.1%)	19 (82.6%)	^†^ 0.025

Min, minutes; IQR, interquartile range. ^†^ Fisher’s exact test/Chi square test. Mann-Whitney U test.

**Table 3 jcm-12-02998-t003:** Predictors of localization accuracy.

Accuracy	Univariant Logistic Regression Analysis	Multivariant Logistic Regression Analysis		
	Odds Ratio	*p* Value	Odds Ratio	*p* Value
	(95% CI)		(95% CI)	
Pre-localization recruitment	0.26	0.024	0.23	0.024
	(0.08–0.84)		(0.06–0.82)	
Lesions	0.61	0.106		
	(0.33–1.11)			
History of lung contralateral wedge resection	0.29	0.039	0.25	0.033
	(0.09–0.94)		(0.07–0.90)	
BMI	0.76	0.001	0.76	0.002
	(0.64–0.89)		(0.64–0.90)	
Ventilation mode	2.43	0.286		
	(0.48–12.41)			

CI, confidence intervals. BMI, Body mass index.

**Table 4 jcm-12-02998-t004:** Predictors of re-inflation.

Re-Inflation	Univariant Regression Analysis		Multivariant Regression Analysis	
	Odds Ratio	*p* Value	Odds Ratio	*p* Value
	(95% CI)		(95% CI)	
Pre-localization recruitment	3.09	0.034	2.99	0.043
	(1.08–8.78)		(1.04–8.64)	
Lesions	1.87	0.048	1.84	0.084
	(1.01–3.47)		(0.97–3.48)	
Previous lung operation	0.43	0.093		
	(0.16–1.16)			
BMI	1.15	0.055		
	(0.99–1.32)			
Ventilation mode	0.31	0.156		
	(0.06–1.57)			

CI, confidence intervals. BMI, Body mass index.

## Data Availability

The datasets analyzed during the current study are not available publicly due to our IRB policy but are available from the corresponding author upon reasonable request.

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
