# Peer review of "Pulmonary Recruitment Prior to Intraoperative Multiple Pulmonary Ground-Glass Nodule Localization Increases the Localization Accuracy—A Retrospective Study"

_jcm, 2023, doi:10.3390/jcm12082998_

Round 1

Reviewer 1 Report

 Thanks to the Authors for giving the opportunity to review this article.

This paper is original and well written.

I have some suggestions :

The definition of "localization accuracy" should be moved from discussion to methods

Authors could be evaluate details of pulmonary function (eg FEV 1) and the possible role on the outcomes

Author Response

Thank you for providing us with positive feedback on our paper. We greatly appreciate your suggestions regarding the manuscript, particularly the importance of including lung function spirometry and diffusion examination in our study. It is worth noting that all patients included in our study were considered ACCP low-risk patients, with FEV1, FVC, and DLCO values above 80% prediction. We have included this additional information in lines 76-79 and have also added the patients' FEV1, FVC, and DLCO values in Table 1: Characteristics of Study Patients. Importantly, we found no statistically significant differences between the recruitment and non-recruitment groups. Thank you again for your valuable feedback and suggestions. The attachment files are the revised manuscript.

Sincerely,

Dr. Yu Hsiang Wang

Reviewer 2 Report

The manuscript entitled “Pulmonary Recruitment Prior to One-Stage Multiple Pulmonary Ground Glass Nodule Localization Increases Localization Accuracy” describes interesting findings related to a very specific anesthesiologic procedure to increase the accuracy of one-stage localization of ground glass nodules possibly improving the feasibility of nodules resection.

Overall, the manuscript is well written, figures and tables are appropriate and the discussion correctly deals with the limitations of such preliminary results.

My main concern is the very specialistic focus of the content. It might benefit of some clarifications, such as:

-       describe the cancer stage of these patients

-       State whether the purpose of the surgery was diagnostic, therapeutic or both. If it was therapeutic and the nodules were > 10 mm or with an emerging solid portion in the same lobe lobectomy would have been the gold standard, not nodule excision. Please discuss.

-       Provide some details (even as a supplementary file) of the pulmonary recruitment technique, to optimize the repeatability of the study

-       State whether or not any other diagnostic assessment was performed (CT guided biopsy)

-       Clarify what is the meaning of “selection bias”, lines 246 – 248. Which are the main features driving surgeon’s/anesthesiologist’s preference of pulmonary recruitment?

-       I would simplify the title. It might be crystal clear for thoracic surgeons but it is not for the majority of clinicians and requires reading the Abstract to be fully understood. Since the Journal is for clinicians and in order to improve its diffusion, I would suggest something like “pulmonary recruitment improves the intra-operative localization of GG nodules”.

Author Response

Dear Reviewer,

Thank you for providing us with positive feedback on our paper. We greatly appreciate your suggestions regarding the manuscript. In Table 1, we have listed the characteristics of the patients with regards to the cancer stage and extent of their surgery. The purpose of surgery in these patients is both diagnostic and therapeutic. However, it remains a concern that if the solid portion (consolidation tumor ratio, C/T ratio) is greater than 0.25 and the total tumor size is greater than 20mm, segmentectomy or lobectomy should be considered as the gold standard. In our study, all patients underwent chest computer tomography prior to their operation, and all patients had C/T ratio measurements. Based on the Japan Clinical Oncology Group (JCOG) 0804 trial, all patients were eligible for wedge resection, which was performed as both diagnostic and therapeutic surgery. All specimens were subjected to frozen section after resection, and if invasive adenocarcinoma or another histological tumor was confirmed, we re-evaluated the lesion size and considered performing lobectomy or segmentectomy as an extensive resection. We have included these details in lines 90-115.

Furthermore, the pulmonary recruitment techniques used were sustained inspiration pressure at 30 cmH2O for 30 seconds by the anesthesia workstation due to its time efficiency and effectiveness. We are planning to conduct another head-to-head study with different recruitment maneuvers later. The details of the recruitment maneuver are listed in lines 101-104 & 253-270.

Regarding the diagnostic assessment, CT-guided biopsy is infeasible for small nodules, and pathologists may find it difficult to evaluate tiny nodule specimens from fine needle aspiration or biopsy.

With regards to the bias of the selection of the pre-procedural recruitment, as our study is retrospective, the pre-procedure pulmonary recruitment was performed based on the preference and experience of the anesthesiologist and surgeon without exact inclusion criteria. While patient characteristics were reviewed, there may have been unmeasured confounding variables that could have affected the results. We have revised the bias in lines 343-357.

As suggested by you and other reviewers, we have changed the title of our paper to "Pulmonary Recruitment Prior to Intraoperative Multiple Pulmonary Ground Glass Nodule Localization Increases Localization Accuracy: A Retrospective Study." Thank you again for your valuable feedback and suggestions. The attachment files are the revised manuscript.

Sincerely,

Dr. Yu Hsiang Wang

Reviewer 3 Report

Evaluation points

1. I suggest inserting the information in the title: Retrospective study;

2. A 1/3 ratio (exposure versus non-exposure) can generate a significant imbalance in the calculation of the OR, suggesting that this observation be considered in the limitations of the study;

3. It would be interesting to add the reasons why the groups were formed with this proportion;

4. The formation of the groups was defined by the anesthesiologist and the surgeon, this implies allocation bias. For this reason, the data must be analyzed with caution and this analysis can also be included in the limitations of the study;

5. The decision criteria regarding the use or not of pulmonary recruitment could be better described so that other investigators have this information as a basis for allocating their volunteers;

6. In Table 01, the abbreviations need a legend or the description in the text must inform the abbreviation right after the first occurrence of the term in the text;

7. The first, second, and third paragraphs of the discussion are written in introductory format and should be revised. Authors can reposition the information leading to the introduction of the article or change the form of the wording so that it can be considered as discussion;

8. The last sentence of the conclusion is disproportionately emphatic when we consider the size of the evidence demonstrated, and for that reason could be softened.

Author Response

Dear Reviewer,

Thank you for sharing your paper with us and for your receptiveness to our feedback. We appreciate your efforts to make revisions to the manuscript based on the suggestions provided by us and other reviewers.

As per the recommendations, the title of the paper has been changed to "Pulmonary Recruitment Prior to Intraoperative Multiple Pulmonary Ground Glass Nodule Localization Increases Localization Accuracy: A Retrospective Study." The proportion of the recruitment and on recruitment group are collected retrospective and generate imbalance between groups. We added this limitation into our discussion in line 356 – 359. Besides, with regards to the bias of the selection of the pre-procedural recruitment, as our study is retrospective, the pre-procedure pulmonary recruitment was performed based on the preference and experience of the anesthesiologist and surgeon without exact inclusion criteria. While patient characteristics were reviewed, there may have been unmeasured confounding variables that could have affected the results. We have revised the bias in lines 343-357, as it was performed based on the preferences and experiences of the anesthesiologist and surgeon without exact inclusion criteria. Although patient characteristics were reviewed, there may have been unmeasured confounding variables that could have influenced the results.

We have also corrected the mistake of the abbreviation in table 01 and he text in the first occurrence of the term, such as FEV1, FVC, DLCO, PCVVG and VCV. Additionally, we revised the wording and content of the first three paragraphs of the discussion to better align with the intended purpose of the section. Finally, we have modified the intensity of the last sentence of the conclusion to avoid misrepresenting the evidence demonstrated in lines 376-384. Thank you again for your valuable feedback and suggestions. Please find the revised manuscript attached.

Sincerely,

Dr. Yu Hsiang Wang